# Individual Concepts in Foot Surgery: A Comparison of Xenogeneic and Autologous Bone Grafts Used in Adults for Lateral Calcaneus-Lengthening Osteotomy According to Evans

**DOI:** 10.3390/jpm13010095

**Published:** 2022-12-30

**Authors:** Markus Siegel, Lisa Bode, Leonard Simon Brandenburg, Andreas Frodl, Hagen Schmal, Jan Kühle

**Affiliations:** 1Department of Orthopedic Surgery and Traumatology, Freiburg University Hospital, Albert-Ludwigs University Freiburg, Hugstetter Straße 55, 79106 Freiburg, Germany; 2Department of Oral and Maxillofacial Surgery, Freiburg University Hospital, Albert-Ludwigs University Freiburg, Hugstetter Straße 55, 79106 Freiburg, Germany; 3Department of Orthopedic Surgery, University Hospital Odense, Sdr. Boulevard 29, 5000 Odense, Denmark

**Keywords:** calcaneus osteotomy, Evans, autologous, xenogeneic, bone graft

## Abstract

Background: Xenogeneic bone grafts, when compared to autologous grafts, are supposed to provide structural benefits without donor site morbidity. To date, there have been divergent results in the use of xenogeneic grafts in foot surgery, primarily in pediatric patient cohorts. The present study examines the incorporation and maintenance of the achieved correction using autologous and xenogeneic bone grafts in adult patients with a six-month follow-up period. Material/Methods: In this retrospective study, 31 adult patients (43 feet in total) treated in our clinic by a lateral calcaneus-lengthening osteotomy, according to Evans, between 01/2006 and 12/2020 were included. The patients were assigned to study groups according to the use of xenogeneic or autologous bone grafts. The osseous incorporation following the criteria of Worth et al., correction maintenance by measuring the talo-navicular coverage angle (TNCA), the talo-first metatarsal angle (TFMA), the calcaneal pitch angle (PCA) and necessary revisions six months after surgery were extracted from the medical files retrospectively. Furthermore, the medical files were screened for the relevant comorbidities, nicotine abuse, BMI, sex and age. Results: In total, 27 autogenous (iliac crest) and 16 xenogeneic bone grafts of bovine origin were used. The evaluation of the radiographs at follow-up demonstrated that there was a mean incorporation rate of 96.3% for the autologous grafts and 57% for the patients treated with xenogeneic grafts (*p* = 0.002). Compared to the autologous group, xenogeneic grafts did not increase the loss of hindfoot alignment in the postoperative course, regardless of being incorporated or not. ΔTNCA, ΔTFMA and ΔPCA displayed no significant differences in both groups (*p* = 0.45, *p* = 0.42 and *p* = 0.10). Conclusion: Despite a significantly lower incorporation rate, the use of xenogeneic grafts was not accompanied with a greater risk of hindfoot alignment loss in the first six months after surgery. Early revision after a postoperative course of six months should not be motivated solely by the radiographic picture of incomplete osseous integration.

## 1. Introduction

The symptomatic pes planovalgus, which results from stage II tibial tendon insufficiency (flexible deformity), is a foot deformity with a high prevalence, afflicting 19.5 to 26% of the world’s population [1,2]. The lateral calcaneus-lengthening osteotomy, according to Evans, is a commonly performed procedure, which is conducted when conservative treatment regimens, such as foot orthotics and exercises, fail [3]. Initially, Evans performed this procedure using an autologous graft placed into an open-wedge osteotomy of the anterior calcaneal process [4]. Several authors modified his procedure [5,6].

Different grafts can be used for the lengthening of the lateral column: autologous, allogeneic, xenogeneic and synthetic grafts with or without a concomitant osteosynthesis were described. Various studies investigated the incorporation of different grafts with or without internal plate fixation, with varying results [7,8,9,10,11,12].

Allogeneic bone grafts were proven to be a good alternative to autologous bone grafts in pediatric patients, in terms of incorporation and hindfoot alignment loss [8,9]. Commonly, radiologic follow-up is performed six to twelve months after surgery. In this period, the bone graft is continuously invaded by migrating osteoblasts but not fully replaced by surrounding bone. Nevertheless, the graft was already exposed to weight-bearing mechanical loads [9].

In terms of correction maintenance and the incorporation rate, good results were also obtained in adult patients. A fundamental advantage of non-autologous grafts is the prevention of donor site morbidity [13,14]. Xenogeneic transplants have been used internationally as an alternative for reconstructive foot surgery for several years, including Evans’ lengthening osteotomy [11]. The proposed benefits of xenogeneic grafts include improved structural integrity and greater mechanical stability [11,15]. The use of xenogeneic bone grafts also seems to be a good way to avoid complications in the context of donor site morbidity and to prevent high costs, at least in comparison to allogeneic grafts [3,16].

Despite these potential advantages, xenogeneic transplants are expected to become incorporated more slowly than allogeneic and autologous bone grafts [17].

Only a few studies investigated xenogeneic bone grafts regarding correction maintenance and incorporation in a lateral calcaneal-lengthening osteotomy. Mainly pediatric patients were examined in this context, yielding divergent results concerning both the incorporation rate and correction maintenance [11,18].

The purpose of this study was to investigate the rate of consolidation and the correction maintenance of autologous and xenogeneic bone grafts in adult patients with pes planovalgus, treated by calcaneus-lengthening osteotomy according to Evans.

We hypothesize that hindfoot alignment loss in the postoperative course at six months does not occur more frequently in patients treated with xenogeneic bone grafts, even if lower incorporation rates compared to autologous bone grafts were reported.

## 2. Materials and Methods

In this retrospective cohort study, a total of 43 feet from 31 different patients were analyzed. These patients were surgically treated by three different foot experts between 01/2006 and 12/2020 in our institution. In total, 23 right and 20 left feet were included, with 12 patients having undergone surgery on both feet in two separate operations.

The study was approved by the Institutional Review Board (the ethics committee of Freiburg University which approved this study, ID 422/18) and registered in the German Trials Register (DRKS00015478). All included patients agreed with study implementation voluntarily, in accordance with the Declaration of Helsinki.

Inclusion criteria were the diagnosis of a radiologically evident pes planovalgus with clinical symptoms (such as pain or general foot fatigue), failed conservative therapy, patient age of 18 to 75, surgery using the modified Evans’ calcaneus-lengthening osteotomy with xenogeneic or autologous bone grafts and a documented postoperative course of therapy with radiographic imaging. A well-documented clinical examination and at least 1 preoperative and 2 postoperative radiographs, including the lateral and dorsoplantar view of the affected foot, were required.

Exclusion criteria were age below 18 or over 75, previous surgery of the investigated foot and other type of material used for the lengthening of the lateral column (such as allogeneic grafts).

The first postoperative radiographic imaging took place during the first two days after surgical treatment to document the postoperative result. This served as a reference point for the further postoperative course.

The primary endpoint of our study was the evaluation of the bony incorporation of the different bone grafts six months after surgery. Secondary endpoints were the postoperative maintenance of correction, the need for consecutive revision surgery and the influence of BMI, sex, age and pre-existing conditions (such as diabetes, vascular diseases and nicotine abuse) on the consolidation process.

A total of 102 records were identified and screened by searching the internal database. In accordance with the inclusion and exclusion criteria, 58 records were excluded. Consequently, 54 procedures could be considered for further analysis. Eight records had to be excluded because no bone graft had been implanted and a further three because the follow-up were not performed in our department. This left 43 procedures which could be included in the analysis. Figure 1 shows a flowchart of the selection process.

In our collective, 27 autologous (AG) and 16 xenogeneic (XG) bone grafts were used. The autologous grafts were taken from the iliac crest. Xenogeneic grafts were of bovine origin (Tutobone^®^, novomedics).

Surgical treatment with a modified lateral lengthening osteotomy according to Evans was performed by one of the surgical foot experts in our institution [6].

The procedure was performed via a lateral calcaneus approach above the peroneal tendons. In addition to the osseous lengthening of the lateral column, an Achilles tendon lengthening was performed according to Strayer, Evans or Young [19]. After surgical treatment with xenogeneic or autologous bone grafts, regular postoperative checks were performed. The majority was fixed by pressfit; in case of doubt regarding stability, plate fixation was added.

In our collective, postoperative immobilization was assured using a plaster cast and patient instruction for partial weight bearing for a total of six weeks.

As part of the six-month postoperative clinical follow-up, native weight-bearing radiological imaging was conducted. The radiologically determined incorporation of the bone graft was evaluated according to the criteria of Worth et al. (see Figure 2) by two independent, blinded investigators [20]. Grades III and IV were considered successful incorporation.

Radiographic foot alignment evaluation was performed by measuring the talo-navicular coverage angle (TNCA), the talo-first metatarsal angle (TFMA) and the calcaneal pitch angle (PCA) in the dorsoplantar and lateral radiographic imaging, preoperatively and in the postoperative course.

For comparison, the patients were divided into 2 groups—the autologous bone graft group (AG) and the xenogeneic bone graft group (XG).

### Statistical Analysis

The descriptive statistics are presented in mean values and standard deviations.

A statistical analysis of the demographic and clinical data was compared using 2-sample t-tests for normally distributed data and a Mann–Whitney rank sum test for not normally distributed data. Categorical variables were compared with Chi-square tests (e.g., gender). Differences in the incorporation rate between the different subgroups were analyzed via Fisher’s exact test.

A logistic regression model was used to identify factors that increase the risk of nonunion.

Due to the retrospective design of our study, no power-analysis was performed before data collection. The calculation uses the IBM SPSS Statistics program, SPSS Statistics 24.0 (IBM Corp., New York, NY, USA). Statistical significance was defined as *p* < 0.05. All analyses were exploratory in nature. As a result, *p*-values and 95% confidence intervals were not corrected for multiple comparisons, and inferences drawn from them may not be reproducible.

## 3. Results

For the analysis, 43 complete data sets were included, of which 27 received autologous (AG) and 16 xenogeneic (XG) bone grafts.

### 3.1. Participants

The patients’ characteristics are found in Table 1. The groups were homogenous in gender distribution and pre-existing conditions, such as vascular diseases (CAD) or diabetes. The AG patients were significantly older and had a lower BMI than the XG patients. The AG patients also showed an increased level of nicotine abuse.

### 3.2. Radiographic Measurements

The first postoperative radiographic imaging, which was performed on the first two postoperative days, showed a good surgical result in all the cases, with a statistically significant correction of the TMFA, TNCA and CPA in both groups (*p* < 0.001) as a starting point for the postoperative course.

Table 2 shows the radiographic angle measurements preoperatively, in the first follow-up and in the second follow-up after a mean time of 6.25 ± 1.27 months. Preoperatively, the groups differed significantly in the lateral TFMA.

In the first follow-up, the TNCA showed a significant difference between both groups. In comparison to the measurements of the second follow-up, there was no significant difference between the two groups looking at the TNCA and the CPA (*p* = 0.06). Comparing the correction maintenance as seen in Table 3, the loss of correction between the first and the second follow-ups showed no significant difference in both groups.

### 3.3. Incorporation Rate

Within six months of the postoperative follow-up, the AG showed a successful incorporation of the transplanted autologous bone grafts in 26 transplants (96.3%). Only one transplant failed to demonstrate sufficient consolidation (3.7%). For the XG at a six-month follow-up, nine transplants (57%) were incorporated. Seven bone grafts (43%) showed grade I or II, according to Worth et al. [20]. Figure 3 gives an example of a successful incorporation (Figure 3A,B) and an example of a failed incorporation with a loss of correction (Figure 3C,D).

When comparing the groups AG and XG in terms of the incorporation rate after six months, the autologous bone grafts were incorporated into the surrounding bone stock significantly (*p* = 0.002) more frequently than the xenogeneic bone grafts.

When comparing the hindfoot alignment of the cases with healed bone grafts (Worth III and IV) after 6 months for the XG group with grafts that had not been fully incorporated (Worth I and II), there are no significant differences in the absolute values or the FU-1 to the FU-2 delta of the respective angles (Delta TFMA: *p* = 0.230; Delta TNCA: *p* = 0.412; and Delta CPA: *p* = 0.239). Table 4 shows the radiographic angle measurements preoperatively in the first and second follow-ups for the XG, divided into cases with incorporated and non-incorporated bone grafts.

### 3.4. Analysis of Patient-Specific and Health-Related Factors

None of the investigated patient-specific and health-related factors were correlated significantly with the consolidation rate (age (*p* = 0.107), the physical constitution in terms of the BMI (*p* = 0.536), the gender of the patients (*p* = 0.285) or the pre-existing conditions, such as nicotine abuse (*p* = 0.811), CAD (*p* = 0.999) or diabetes mellitus (*p* = 1.000)).

Overall, within our follow-up period, one subject (6.2%) of the XG with insufficient incorporation, a significant loss of correction (TFMA−10.2 degree) and clinical symptoms (strong pain) in the postoperative course received revision surgery. Another revision surgery for the XG was conducted due to a symptomatic bony nonunion 21 weeks after surgery, without a significant loss of correction. Both subjects underwent revision surgery with autologous bone grafts.

Within the first six months, we had a complication rate and a revision rate of 12.5% for the XG group, while there were neither relevant complications nor revisions in the AG group (*p* = 0.16).

## 4. Discussion

The main findings of the present study are that xenogeneic bone grafts have a significantly lower incorporation rate than autologous grafts six months after surgery. Nevertheless, there was no increased loss of hindfoot alignment in the XG compared to the AG. Moreover, within the XG group, there was no difference in the correction maintenance of the healed versus non-healed bone grafts. The factors sex, age, BMI, nicotine abuse, CAD and diabetes mellitus had no significant influence on the incorporation rate. Although there were two patients undergoing revision surgery in the XG and zero in the AG group, this difference did not reach significance.

The use of xenogeneic bone grafts in foot surgery has been the subject of intense discussion [11,17,18]. While the physical properties of xenogeneic grafts promise good surgical results, previous studies have shown delayed incorporation rates for xenogeneic transplants used in reconstructive foot surgery [15,17].

The results of our study show that, after a mean follow-up of 6.25 months, only about half (57%) of the xenogeneic grafts were incorporated.

Comparable to the chosen study design by Müller et al. (2016), we restricted our follow-up period from directly postoperative to 9.5 months [9]. During this time period, the transplanted graft is continuously invaded and mechanically loaded. In early postoperative follow-up care, a correction loss is closely associated with the failure or collapse of the bone graft rather than the secondary loss of correction caused by the degenerative processes of tendons or ligaments [9].

The presented incorporation rate is comparable to the results obtained by Rhodes et al. in a pediatric patient population. In this study, a prolonged incorporation rate for xenogeneic transplants of 29% after 6 months and 73% after 9 months in comparison to allogeneic bone grafts was detected. Though, their criterion for successful incorporation was Grade IV according to Worth et al. [11,20].

The poor integration of xenogeneic transplants in reconstructive foot surgery was also shown in other studies [17,18]. In 2013, Ledford et al. showed a similar incorporation rate to our findings by investigating 13 operations with xenogeneic grafts in pediatric patients with foot deformities. In their study, seven patients (53%) showed clinical symptoms of failure with a corresponding radiographically failed graft incorporation. This resulted in revision surgery in all seven cases, with allografts used for the revision [18]. However, the comparison of these studies to our results is lacking due to the pediatric patient population. In an evaluation by Shibuya et al., who investigated various reconstructive foot surgery procedures in which xenogeneic grafts were used, an incorporation rate of only 22.2% (two of nine) after six months was found for subjects treated with an Evans osteotomy [17]. In this study, an age distribution of 9 to 80 years was used for all the patients included.

For autologous bone grafts, the research has shown very good results for the incorporation rate within the first 3–6 months, with up to 100% successful incorporation [4,13,14]. The same can be seen in our population, with an incorporation rate of 96% after about six months.

In our study, a difference between the AG and XG groups showed a significantly younger average age, less tobacco use and a higher BMI for the XG group. While a younger age is considered a positive predisposing factor for faster bone healing, the opposite is true for tobacco use and a higher BMI [21]. However, there was no significant correlation between the incorporation rate and the investigated patient-specific and health-related factors extracted from the medical charts.

Despite the significantly poorer incorporation rate of xenogeneic bone grafts, there was a similar maintenance of the correction in the postoperative course of six months when considering the angles we chose for the evaluation (TFMA, TNCA and CPA) in comparing the two groups. For the evaluation of the hindfoot alignment, previous studies have shown that the TMFA is a sensitive marker for plantar pressure distribution under dynamic loads when compared to other angles [9,22]. Though the TFMA differed preoperatively in both groups, a comparable maintenance of the correction in both groups was shown in our study after surgical treatment in the six-month follow-up.

There are no significant differences between the correction maintenance of all the angles investigated between the incompletely incorporated bone grafts and the healed bone grafts within the XG group. This leads us to the assumption that, despite a slower incorporation of the xenogeneic when compared to the autologous grafts in adult Evans surgery, a loss of correction is not inevitable. Early revision after a postoperative course of six months should therefore not be motivated solely by the radiographic picture of incomplete osseous integration.

There are no statistical differences between the two groups when comparing the complication and revision rates. The possibility that delayed incorporation leads to further complications and possibly to revision surgery in the further postoperative course is certainly given but could not be ascertained in our study.

To point out the advantages of xenogeneic bone grafts, as with allogeneic or synthetic grafts, there are no donor site morbidities, such as wound infections, pain, neuralgia paresthetica or iliac wing fractures due to bone harvesting [14,23]. Though there were no reported donor site complications in this study, several studies have shown a major complication rate (between 0 and 11.5%) in harvesting autologous bone grafts from the iliac crest [16]. This suggests that the use of autologous bone grafts might be dependent on the surgical technique and experience. Using xenogeneic grafts could avoid such severe complications.

Another aspect to be considered in the choice of bone graft is the cost. As shown in several studies, another argument for xenogeneic bone grafts is that they are less expensive than allogeneic grafts [24].

To summarize the results of this study, we can state that autologous grafts appear to be superior to xenogeneic transplants in terms of the osseus incorporation within the first six months after surgery in adult patients treated by a calcaneus-lengthening osteotomy according to Evans. Despite this finding, there was no significant difference in the maintenance of correction in both groups. In our collective, there was no statistical relevance for the factors age, sex or BMI, nor for tobacco use, CAD or diabetes on the consolidation rate.

## 5. Limitations

The retrospective design prevented the standardization of the group size, as well as in terms of age and sex matching. Furthermore, we did not perform a power-analysis prior to the study as the group size was predefined by the retrospective design of the study. The rather short follow-up period of our study was due to our postoperative protocol, which scheduled a postoperative clinical examination and radiological control after six months in order to reveal complications (such as a loss of corrections and pseudarthrosis). One aim of this study was to reveal the causes of early failure. In this regard, the follow-up time of six months is considered appropriate for attributing complications predominantly to the osseous integration of the bone graft. The groups differed significantly in age, BMI and nicotine abuse. Nonetheless, there was no statistical influence of patient age or BMI with the incorporation rate in our study. Thus, the results of the AG and the XG groups could be compared, at least in the early postoperative course. Moreover, the AG patients were significantly older than the XG patients but displayed a significantly higher rate of consolidation after a six-month follow-up, which underlines the superiority of autologous bone grafts regarding incorporation. A larger collective might show divergent results. The study did not include patient-related outcome measures. Therefore, it cannot be evaluated whether the lacking bony integration was associated with more pain or other complaints.

## 6. Conclusions

Despite a significantly lower incorporation rate, xenogeneic grafts do not entail a higher risk of hindfoot alignment loss within the first six months after surgery. Thus, xenogeneic grafts prove to be an adequate alternative, without causing any donor site morbidities. Early revision after a postoperative course of six months should not be motivated solely by the radiographic picture of incomplete osseous integration.

## Figures and Tables

**Figure 1 jpm-13-00095-f001:**
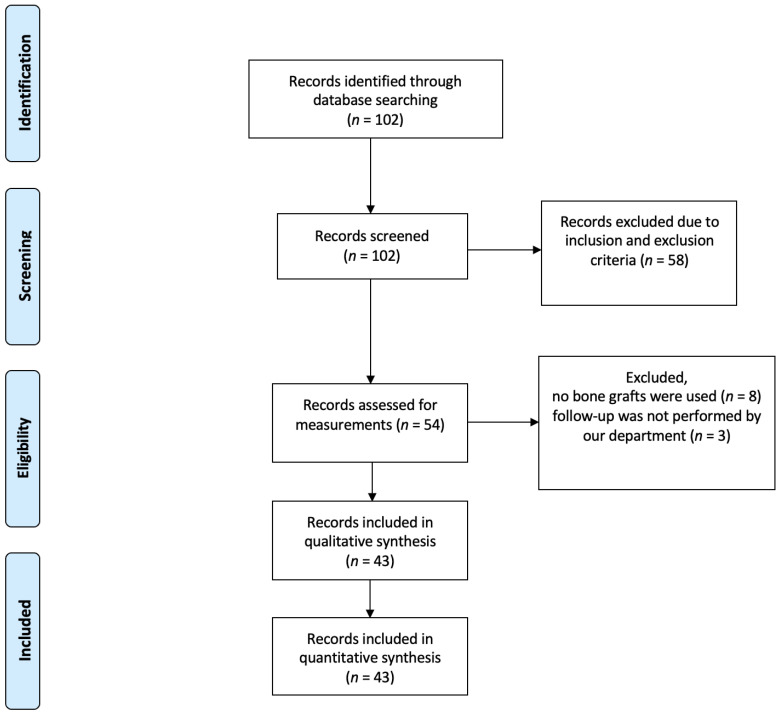
Flowchart of the selection process.

**Figure 2 jpm-13-00095-f002:**
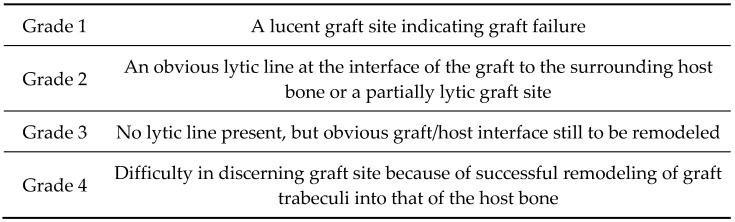
Adapted and reprinted grades of incorporation by Worth et al., 2005 [20].

**Figure 3 jpm-13-00095-f003:**
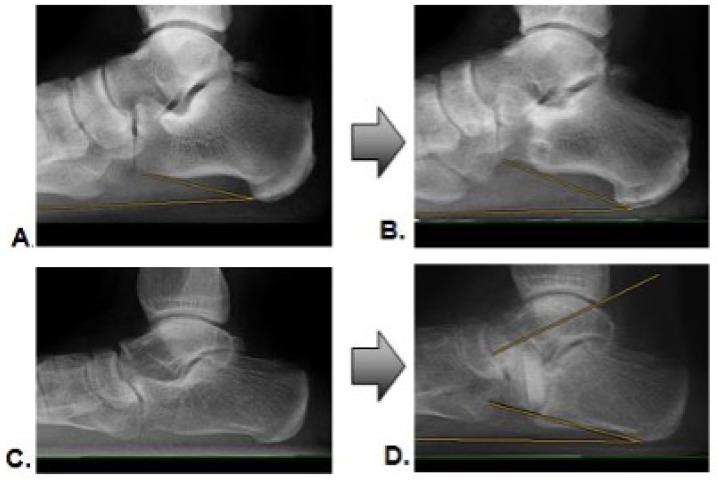
Radiographic example of pre- and postoperative images with successful incorporation (**A**,**B**), and loss of correction with incomplete incorporation (**C**,**D**).

**Table 1 jpm-13-00095-t001:** Patient characteristics including gender, age, body mass index (BMI) and health-related factors: nicotine, diabetes and coronary artery disease (CAD); * Chi-square test, ^#^ Mann–Whitney rank sum test, ^+^ Student’s *t*-test.

Characteristics	All Patients	Autograft (*n* = 27)	Xenograft (*n* = 16)	*p*-Value
Gender				
Male sex (*n*)	25	16	9	0.548 *
Female sex (*n*)	18	11	7	
Age (yr)				
Mean	34.43 ± 17.25	40.1 ± 17.58	23.4 ± 9.48	<0.001 ^#^
Range	18 to 75	20 to 75	18 to 58 years	
BMI (kg/sqm)				
Mean	25.51 ± 4.70	22.98 ± 3.9	26.99 ± 4.5	0.003 ^+^
Range	17.6 to 36.4	17.6 to 31.0	21.19 to 36.4	
Nicotine (*n*)	9	8	1	0.038 ^#^
Diabetes (*n*)	1	1	0	0.327 ^#^
CAD (*n*)	2	2	0	0.161 ^#^

**Table 2 jpm-13-00095-t002:** Angle measurements preoperatively and in the postoperative course of the TFMA, TNCA and CPA. * TFMA negative values equal plantar convexity, † time in months, ’ angles in degree.

Parameter	All Patients	Xenograft	Autograft	*p*-Value
Preoperative				
Mean Preop. Time (range) ^†^	4.07 (0.1 to 16.0)	3.67 (0.2 to 11)	4.29 (0.1 to 16)	0.58
Mean TFMA * (range) ’	−13.96 (−37.7 to 14.5)	−9.94 (−32.6 to 0.8)	−16.34 (−37.7 to 14.5)	0.02
Mean TNCA (range)	18.24 (−12.3 to 46.6)	16.43 (2.1 to 27.4)	19.3 (−12.3 to 46.6)	0.32
Mean CPA (range)	16.65 (8.3 to 29.4)	15.44 (8.3 to 21.1)	17.37 (9.0 to 29.4)	0.18
1. Follow-up (FU-1)				
Mean TFMA (range)	−0.31 (−16.8 to 13.7)	−1.44 (−16.8 to 13.7)	0.34 (−11,2 to 9.3)	0.40
Mean TNCA (range)	1.57 (−20.7 to 18.4)	−2.66 (−20.7 to 9.8)	3.84 (−12.2 to 18.4)	0.04
Mean CPA (range)	24.81 (14.7 to 32.1)	23.85 (14.7 to 31.5)	25.37 (16.2 to 32.1)	0.36
2. Follow-up (FU-2)				
Mean FU Duration (range)	6.25 (4.5 to 9.5)	6.33 (4.5 to 8.0)	6.20 (4.5 to 9.5)	0.73
Mean TFMA (range)	−4.25 (−20.8 to 11.3)	−3.91 (−16.1 to 11.3)	−4.47 (−20.8 to 4.5)	0.78
Mean TNCA (range)	3.81 (−21.1 to 22.4)	0.64 (−21.2 to 13.5)	6.01 (−6.7 to 22.4)	0.06
Mean CPA (range)	20.83 (10.6 to 32.6)	19.14 (11.8 to 30.7)	21.96 (10.6 to 32.6)	0.06

**Table 3 jpm-13-00095-t003:** Delta of the angles measured between follow-up (FU) 1 and follow-up 2.

Parameter	All Patients	Xenograft	Autograft	*p*-Value
Delta mean FU-1 to FU-2				
Delta mean TFMA (range)	−3.94 (−10.2 to 4)	−2.47 (−10.2 to 4)	−4.81 (−9.7 to 2.2)	0.42
Delta mean TNCA (range)	2.24 (−13.5 to 16.6)	3.30 (−2.8 to 16.6)	2.17 (−13.5 to 15.6)	0.45
Delta mean CPA (range)	−3.98 (−12.3 to 0.5)	−4.71 (−12.3 to 0.3)	−3.41 (−6.5 to 0.5)	0.10

**Table 4 jpm-13-00095-t004:** Angle measurements for the XG with incorporated bone grafts after the FU-2 and unincorporated bone grafts after the second FU-2 preoperatively, then in the postoperative course of the TFMA, TNCA and the CPA; ’ angles in degree.

Parameter	Incorporated Bone Grafts XG	Incomplete Incorporated Bone Grafts XG	*p*-Value
Preoperative			
Mean TFMA (range) ’	−14.80 (−32.4 to −5.7)	−16.33 (−26.6 to −5.6)	0.366
Mean TNCA (range)	14.72 (2.1 to 23.9)	18.63 (7.9 to 27.4)	0.154
Mean CPA (range)	16.13 (11.8 to 20.4)	14.54 (8.3 to 21.1)	0.234
1. Follow-up (FU-1)			
Mean TFMA (range)	5.31 (0.4 to 10.4)	−0.28 (−12.6 to 16.7)	0.117
Mean TNCA (range)	−3.60 (−20.7 to 5.1)	−1.40 (−18.4 to 9.8)	0.347
Mean CPA (range)	24.06 (14.7 to 31.2)	23.64 (17.0 to 31.5)	0.440
2. Follow-up (FU-2)			
Mean TFMA (range)	0.66 (−10.3 to 12.6)	−1.89 (−15.4 to 15.6)	0.285
Mean TNCA (range)	−0.03 (−10.9 to 9.2)	1.51 (−21.2 to 13.5)	0.384
Mean CPA (range)	18.81 (13.9 to 23.2)	19.56 (11.8 to 30.7)	0.387

## Data Availability

All data are within the manuscript.

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
