# Peer review of "Individual Concepts in Foot Surgery: A Comparison of Xenogeneic and Autologous Bone Grafts Used in Adults for Lateral Calcaneus-Lengthening Osteotomy According to Evans"

_jpm, 2022, doi:10.3390/jpm13010095_

Round 1
Reviewer 1 Report
The present paper is well written, providing all necessary informations.
The subject is of actual interest due to the need of surgical treatment for the chronic or trauma patients.
I hope you will continue this study with prolonged period of follow-up and more patients.
Author Response
Dear Reviewer 1,
Thank you very much for your positive review, I was very happy about it.
Yours sincerely,
Markus Siegel
Reviewer 2 Report
The idea of the authors of the manuscript is very interesting and the study is well approached.
It needs a minor stylistic revision before being suitable for publication:
1. The abstract appears very long, try to summarise it, avoiding repetition (consider a maximum of 250 words)
2. In table 1 specify the abbreviations BMI and CAD
3. Table 1 should be moved to the results and not to the materials and methods section
4. Once the abbreviations have been specified, it is no further need to write them in full (e.g. see line 179 with TFMA, correct the text accordingly).
5. Line 235-236, in addition to citations 11-17-18, if the authors consider it appropriate they could cite another foot condition that is a source of access bone grafting debate (PMID: 30644297).
Author Response
Dear Reviewer number 2,
thank you for your positive review.
1. The abstract appears very long, try to summarise it, avoiding repetition (consider a maximum of 250 words)
We have significantly revised the abstract again and summarized it even more. We hope that it is now more concise and substantial.
2. In table 1 specify the abbreviations BMI and CAD
We have now specified the abbreviations BMI and CAD.
3. Table 1 should be moved to the results and not to the materials and methods section
We have transferred Table 1 and the short associated paragraph to the Results section.
4. Once the abbreviations have been specified, it is no further need to write them in full (e.g. see line 179 with TFMA, correct the text accordingly).
We have adjusted the abbreviations already specified in various places in the text.
5. Line 235-236, in addition to citations 11-17-18, if the authors consider it appropriate they could cite another foot condition that is a source of access bone grafting debate (PMID: 30644297).
This is a very detailed paper. However, after reading through the suggested source, I could not find any discussion of xenogeneic bone grafts. Although I think the suggested source is a very interesting publication, I do not think it is appropriate to cite the suggested study here.
many thanks,
Markus Siegel